# Recent Status and Prospects on Thermochemical Heat Storage Processes and Applications

**DOI:** 10.3390/e23080953

**Published:** 2021-07-26

**Authors:** Tadagbe Roger Sylvanus Gbenou, Armand Fopah-Lele, Kejian Wang

**Affiliations:** 1College of Mechanical and Electrical Engineering, Beijing University of Chemical Technology, Beijing 100029, China; 2019420018@mail.buct.edu.cn; 2Department of Mechanical Engineering, Faculty of Engineering and Technology, University of Buea, Buea P.O. Box 63, Cameroon; a.fopah-lele@ubuea.cm

**Keywords:** thermochemical heat storage, reactor, thermal simulation, heat storage application

## Abstract

Recent contributions to thermochemical heat storage (TCHS) technology have been reviewed and have revealed that there are four main branches whose mastery could significantly contribute to the field. These are the control of the processes to store or release heat, a perfect understanding and designing of the materials used for each storage process, the good sizing of the reactor, and the mastery of the whole system connected to design an efficient system. The above-mentioned fields constitute a very complex area of investigation, and most of the works focus on one of the branches to deepen their research. For this purpose, significant contributions have been and continue to be made. However, the technology is still not mature, and, up to now, no definitive, efficient, autonomous, practical, and commercial TCHS device is available. This paper highlights several issues that impede the maturity of the technology. These are the limited number of research works dedicated to the topic, the simulation results that are too illusory and impossible to implement in real prototypes, the incomplete analysis of the proposed works (simulation works without experimentation or experimentations without prior simulation study), and the endless problem of heat and mass transfer limitation. This paper provides insights and recommendations to better analyze and solve the problems that still challenge the technology.

## 1. Introduction

Global energy demand continues to increase with high growth in the exploitation of fossils fuels, whose prices and environmental issues are leading scientists and engineers to find inclusive solutions. The use of renewable energy fits these challenges well. Solar energy, mainly known for its infinite and renewable nature, is one of the most widely used today. Due to its seasonal availability, the major challenge continues to be the need to store solar energy in periods of high availability to be re-used in periods of shortage. Several solar energy storage methods have been developed, among which TCHS appears to be one of the most promising. TCHS processes have the potential to store heat over theoretically infinite time and long-distance transportation. More recently, an increasing number of laboratory-scale as well as large-scale projects are being developed in the field of energy storage for both low-temperature and high-temperature applications, but the technology is still not fully accessible and marketable. To further identify the problems and limits encountered, several synthesis works are published, reviewing the progress of the studies and mentioning the future challenges to be tackled [1,2,3]. Desai et al. [4], in their recent review on TCHS systems for both heating and cooling process applications, have summarized some of these works with their highlights, as shown in Table 1.

Many of these works have shown promising results in terms of convertibility of the energy stored by the material, the development of innovative systems that are compact and simple to maintain, the availability of output data and their analysis, and the improvement of the efficiency of the existing system. However, despite the advantages offered by the process, the perspicacity of their scientific approach, and their achievements, they have only reduced the effects of the problems encountered, and several drawbacks remain: the costly investments, the high cost of the storage materials, the limited heat and mass transfer capacity, and the poor heat storage density in the current systems. This review provides a recent state of the art of TCHS technologies, including some key terms and concepts, advances in TCHS systems, and an analysis of some relevant recent research. The most significant findings and perspectives are provided.

## 2. General Concepts of TCHS

Heat is the predominant form of energy used in industry, in the building heating system, and in the power plants to run the processes or to generate electricity. In proportion to the work produced, a significant amount of this heat is lost to the environment through cooling water from cooling towers, thermal insulation, or other ways. Therefore, because of its relatively low quality, the exhaust heat is neglected. The TCHS offers the possibility of reusing the waste heat by recycling and upgrading it [7,14,15,16,17,18,19,20]. In addition, when the heat request is located at a certain distance from the supply, this heat can be conveyed. Thus, it considerably reduces the delay between the supply and demand of energy. TCHS can also enhance the efficiency of using the solar energy potential [21]. Similarly, in diesel or gasoline engines, TCHS may be used to recycle the waste heat from the radiator or exhaust. Typically, in vehicles, this waste heat represents 60% of the fuel energy [22,23,24,25], and in reality, only 20% of the fuel energy is used to power the vehicle [26,27]. As a result, recycling this heat and reusing it for heating or cooling applications would reduce fuel consumption and significantly improve the performance of the engine. Thermal energy can be stored as sensible heat, latent heat, or as chemical potential by TCHS processes, which include the sorption process (absorption and adsorption) and chemical reaction [28]. Among these, the storage process by adsorption and chemical reactions has the greatest potential for energy efficiency, energy savings, and minimizing the emissions of CO_2_ [29,30,31,32,33,34]. In both of these cases, the heat is stored directly as a chemical potential by a reversible chemical process or reaction. However, up-to-date, thermochemical heat storages remain largely in the research and development (R&D) stage. Concerning systems involving chemical reactions, 95% of installed systems are in R&D and have reached a technology readiness level (TRL) of 3–4. Sorption storage systems are slightly more developed (TRL 5–7), except for sorption heat pumps, which have been fully marketed (TRL 9) [35,36]. One of the main reasons for the different TRLs is directly related to the materials cost/performance and indirectly to reactor design. However, despite the performance of such a system, the two old thermal storage processes (sensible and latent heat storage) are still the most widely used, and the few prototypes developed in TCHS are still at the laboratory and test stage [37]. It is, therefore, necessary to redefine the framework and purpose of thermochemical heat storage to properly address the barriers that still hinder research. To raise the TRL of TCHS and create a commercialized device, attention needs to be redirected from material (or microscopic) issues to reactor (or macroscopic) issues.

## 3. The Different Processes for TCHS

TCHS is a recent energy storage process offering the benefit of very high-energy storage densities and very low heat losses during the process of storing and transporting the energy. These characteristics give the process a very attractive and advantageous long-term energy storage at both low and high temperatures. The process is characterized by a reversible reaction between reactants and products involving a process of endothermic decomposition and exothermic synthesis.
X(s) + Y(g) ↔ XY(s) + heat.(1)

Compounds X and Y may be stored separately when the energy is stored, depending on the nature of the compounds and the chemical reactions they involve [38,39]. This storage can be in different forms: either relative to physisorption in adsorption grids (adsorption) or relative to absorption in absorbent solutions or also relative to chemical reactions as chemical potential. Applications such as building space heating, domestic hot water supply, or sterilization of certain work tools may involve the use of sorption heat storage processes while high-temperature heat uses such as in gasification process, the load shifting, the power generation, or electricity production processes may require chemical heat storage using pure chemical reactions to store energy as chemical potentials. Through the literature [40], TCHS processes can be organized as shown in the following diagram in Figure 1.

In addition, before deciding on the appropriate process to be used for storing energy, it is recommended that an in-depth analysis should be carried out on a selected number of system parameters. Among these parameters, the following can be quoted:Storage capacity: this defines how much energy the system can store and varies significantly according to the process, the device, and the scale of the system involved.Power: it defines the speed at which the energy stored in the system can be released (and loaded).Efficiency: it is the balance between the energy received by the user and the energy required to charge the storage unit. It takes into account the energy losses during the storing process.Storage period: this is the total time that energy is stored, ranging from few hours to several months. This enables characteristics such as short- or long-term storage.Charge/discharge time: it defines the duration needed to fully charge and discharge the system.Cost of the technology: it evaluates the cost price of the system by referring either to the capacity (cost/kWh) or the power (cost/kW) of the system and fluctuates according to the investment and operating costs of the equipment involved and their life span.

### 3.1. Absorption Storage Process

This process usually occurs between liquid and gas, leading to the formation of a new component. The absorption phenomenon can be defined as the physical or chemical process by which molecules of a substance penetrate the layer of a liquid or solid surface by integrating the structure of the solid or liquid, resulting in a change in the composition of the structure. When the absorbent is a liquid and the absorbate is a gas, the process can be described as liquid–gas absorption [41]. Scientific advances in absorption storage processes have focused more on cooling and air-conditioning systems. 

Concerning the absorption technologies developed, absorption refrigeration is the most commonly implemented technology for solar cooling [42,43,44,45,46,47,48,49,50,51,52,53,54,55,56,57,58,59,60,61,62,63,64,65,66]. It involves very low or no electricity consumption, and for an equal cooling capacity, the size of an absorption refrigeration unit is mostly undersized compared to other units, owing to the significant amount of heat and mass transfer. The principle of the absorption system is illustrated in Figure 2.

In the summer period, while charging the system, the diluted absorbate solution is directed to the desorbed. Under the effect of the heat brought by the solar collector, the solution releases the absorbate vapor and becomes concentrated in absorbent, which is stored in a concentrated solution tank. Charged with heat, the released vapor is then stored in a liquid form in the absorption vessel after passing through the condenser via a heat sink on which it is condensed. To avoid mass transfer across the components, the storage vessels are disconnected from each other as soon as the charging period has elapsed. During the winter, the discharging phase can occur. The absorbing material is evaporated in the absorber as the heat is transferred through the evaporator from a source at a lower temperature. Once in the absorber, it is then be soaked up into the concentrated solution flowing out of the storage reservoir. This absorption phenomenon is exothermic and releases heat, which can be used for building heating purposes or for other applications. Despite the maturity of the system, many challenges such as storage lifetime, recycling of materials, crystallization problems in the exchanger columns, and the lifetime of the materials remain to be met.

### 3.2. Adsorption Storage Process

Adsorption can be generally defined as a surface phenomenon by which gas or liquid molecules adhere to the solid surfaces of adsorbents. The molecules thus adsorbed constitute the adsorbate. Adsorption is usually based on the property that solid surfaces have of fixing certain molecules reversibly, by weak Van der Waals-type bonds. More precisely in physics, this property is linked to the structure of the solid itself, where unbalanced forces remain at the surface due to the unequal distribution of the atoms: the formation of a layer of adsorbed molecules partially compensates for this imbalance [67]. According to the nature of the boundary between the adsorbent and the adsorbate, the adsorption phenomenon can be grouped into four types: solid/gas, solid/liquid, liquid/liquid, and liquid/gas. Among the mentioned adsorption types, solid/gas adsorption is the one that has received extensive investigation and research, contributing significantly to the interface chemistry. Hence, when referring to adsorption or solid adsorption, the term solid/gas adsorption is generally used, and according to the binding force between the two phases, adsorption is separated into two forms: physical adsorption (physisorption) and chemical adsorption (chemisorption) [68]. The primary interaction force of physisorption is Van der Waals, and it can be defined as a type of adsorption that occurs for most particles in contact with a solid or liquid surface. Condensation of water molecules sticking to a drinking glass is an example [69]. Chemical adsorption is distinguished by chemical interactions that can fluctuate from very low values to a value at least 100 times higher than physisorption using covalent forces similar to those occurring in the chemical compound formation. There are certain differences in the properties of the two kinds of adsorption, which can be used as experimental criteria for deciding the adsorption type. The ideal way to compare these two forms of adsorption is to use the scale of the heat of adsorption involved [13]. In practice, the differentiation between a particular adsorption as physisorption or chemisorption depends mainly on the binding energy of the adsorbate to the substrate, with physisorption being much lower per atom than any type of connection involving chemical bonding. In general, the principle of storage by adsorption can be explained through Figure 3.

In Figure 3, zeolite and water are used, respectively, as the adsorbent material and the adsorbate to achieve adsorption storage. When charging the storage unit, with the input of heat $Q_{des}$ l, the storage unit temperature rises, and the desorption process takes place. Steam is freed from the zeolite to be condensed at a lower temperature. As a result of the condensation, the heat generated, $Q_{cond}$, can either be recycled in the process or discharged into the immediate environment. A reverse process occurs when the storage unit is discharged: the heat of poor rank $Q_{evap}$ is applied to evaporate the water. Hereby, the emitted water vapor is therefore adsorbed by the zeolite, and the released enthalpy of adsorption $Q_{ads}$ can serve as a heat source in the process where a higher temperature is required. Until now, the value of the adsorption rate (quantity of adsorbate adsorbed by the adsorbent by a unit of mass or volume) achieved by certain studies and experiments carried out is not yet satisfactory [70,71,72,73,74,75,76,77,78,79,80,81,82,83,84,85]. Moreover, maturity and self-sufficiency are still the major challenges to be addressed. Research work continues to be developed to overcome those limits.

### 3.3. Chemical Reactions Storage Process

For power generation applications requiring both high working temperatures and high enthalpies of reaction, chemical reaction storage is preferred. In the literature, the chemical reactions concerned are of two types: the coordination reaction of ammonia and the hydration reaction of salt hydrate with water [86,87]. The issue of classifying the coordination reaction as solid sorption or a chemical reaction remains unclear. Wang et al. [68] and N’Tsoukpoe et al. [88] were inclined to use chemical adsorption (chemisorption) to describe the coordination reactions. The reason is possibly due to the importance of the part played by the solid surface in these solid/gas reactions, while Cot Gores et al. [7] have identified these reactions as solid/gas absorption reactions due to the molecular structure transformation that occurred. To avoid possible contestation and ambiguity, the concept of chemical reaction appears to be an acceptable approach [89] and therefore will be adopted here. The reactions occurring are reversible. The chemical reaction involved is as follows:AB + Q ↔ A + B(2)

During the decomposition/dissociation phase of the product AB, the thermal energy Q enables the endothermic reaction to take place. The products A and B are then stored as chemical potentials. This is the energy charging stage. The reversible reaction is formed exothermically and refers to the formation/restitution step. The products A and B are brought into contact to regenerate the initial product AB and release a quantity of heat Q; this is the energy discharge stage. Figure 4 illustrates the described process.

Chemical heat storage offers several advantages over the other two TCHS processes [90,91,92,93,94,95,96,97,98,99,100,101,102,103,104,105,106,107]:The energy density is, respectively, higher than for adsorption and absorption storage systems.The temperature can rise to 1000 °C for a certain application.The heat can be restored at a constant temperature.The storage time, as well as the transport distance of the reagents, is theoretically unlimited since the products are stored under ambient temperature. This process is the most suitable heat storage process for seasonal storage, i.e., storing energy in the summer and releasing it in the winter for a long duration.

It is important to mention that the complexity involved in the technology as well as the bad integration between the heat sources and the heat sinks with a long load time-limited by mass transfer in the system remain major challenges.

To overcome the constraints of increasing the capacity of the materials used for pure chemical reactions, a promising new process is being developed and continues to receive improvement and scientific advancement. These are chemical reactions with composite materials [108]. This process is similar to the previous one, but here, the mixture of known standard materials is used with a new material that has specific characteristics for improving heat and mass transfer properties and increasing the durability as well as the stability of such composite material over time [109,110,111].

Research on the development of composite materials is a recent known field of thermochemical heat storage to enhance the heat transfer in thermochemical reactors, and many research structures are interested in it.

## 4. TCHS Materials

To perform TCHS, we need materials that can store and release heat at any time through specific chemical reactions when needed. Such materials offer the benefits of high density for storing the heat, a considerable raising of the temperature, and the ability to store the reagents (sorbent and sorbate) at room temperature without auto-discharge, which is an important feature for applications in seasonal storage. Referring to [112,113,114,115], TCHS materials should meet many criteria as follows: The high affinity of the sorbent for the sorbate: impacts the rate of the reaction.Better volatility of the sorbate than the sorbent in absorption.High thermal conductivity and high heat transfer with the heat transfer fluid in the case adsorption.Desorption temperature as low as possible and suitable permeability.Environmental safety, non-toxicity, low global warming potential and ozone depletion potential, and low cost.Non-corrosiveness of materials and a low recovery temperature to ensure high solar fractions.Good thermal and molecular stability under assigned operating conditions (temperature, pressure).Moderate operating pressure range, no excessive pressure conditions, and especially no high vacuum.

Usually, analyses are performed using differential scanning calorimetry (DSC), using the thermogravimetric analysis tool (TGA) to establish the thermal behavior of the materials. Various types of TCHS materials have been identified as suitable for application in energy storage systems, but just a few have been successfully introduced in pilot and bench-scale applications [116,117,118,119,120]. In this review, a summary and classification of these materials according to the storage processes is made, the most interesting are discussed, and some solutions to improve their properties are proposed. 

### 4.1. Materials for Absorption and Adsorption Storage Processes

The materials used for ad/absorption phenomena are appropriate to store heat for applications in building space heating, domestic hot water applications, and more generally in processes requiring the supply of heat at low temperatures. They involve liquid/gas (absorption) or solid/gas (adsorption) reaction. Sorption materials have been mostly investigated in the field of TCHS. Some of them exhibited a very promising potential for storing energy. Investigations on sorption storage material characterization are still conducted to design efficient TCHS materials.

Table 2 gives the details on the type and characteristics of some sorbents, and Table 3 shows candidate materials for special application in building space heating.

It is important to note that in absorption storage systems, storage energy density is strongly related to the concentration of the storage solution. In a study conducted by Liu et al. [128] on several materials, the candidate material LiBr/H_2_O appears to be the best-performing material in terms of effective storage capacity and storage efficiency, but its low availability due to its high price could hinder its application in seasonal energy storage systems regarding the cost of the technology. To overcome this problem of high costs, CaCl_2_/H_2_O appears to be the most suitable candidate material. However, it has a very low storage capacity. A new liquid storage material must be developed to enhance the heat exchange in the system as well as the storage density. Some laboratories, such as TREFLE at the Mechanical and Engineering Institute of Bordeaux in France, have developed liquid storage materials called sugar alcohol, which have a very high energy density and are capable of being stored at room temperature to contribute to the European projects for TCHS [116]. These materials in the shape of crystals that melt at a temperature of about 70–90 °C therefore store energy at this temperature, and this energy is recovered by a bubbling process, i.e., air is injected into this liquid to force crystallization by creating a bubble column to release the energy at this time. The difficulty with these materials is that they are very viscous, and this limits the rate of crystal growth. This can have the advantage of not losing at any time for the stored energy but generates the problem of mass transfer to the system’s inside.

### 4.2. Materials for Chemical Reaction Energy Storage Process

Before delving into this section, it should be noted, as previously announced, that there is no precise distinction between the chemical reactions with sorption. It is generally classified as a sorption process or a chemical reaction that does not involve sorption. However, compared to liquid absorption and solid adsorption, chemical reactions are mono-variant and lead to transformation about the volume of the solid [129,130,131,132,133]. The main chemical reaction storage materials developed and studied can be grouped into carbonate decomposition material, redox material, inorganic hydroxide material, ammonia decomposition material, metal hydride material, and methane reforming material. Table 4 presents a summary of these materials, studied recently, as well as their characteristics and their limits.

As far as composite materials are concerned, the authors describe them as the result of the mixture of salt hydrated material and a porous structure material with a high thermal conductivity [110]. Composite materials are required for specific applications such as some specific industrial processes working at high temperatures. It can be noted that the recent developments in material science for energy storage are very promising and offer the possibility to increase storage capacities when the optimal combination is reached. Some of them are more promising than others. Advances in the field have led to the discovery of new approaches combining materials with different properties to improve the quality of storage or to define a use for specific applications such as heat production and others. Details about the design methods of composite materials can be found in [134]. However, the high cost of the technologies used to manufacture the materials, the inefficient recycling, the toxicity, the high cost of the raw materials, and the limited heat transfer in porous materials are so many problems that affect this sector. The actual use of these materials in practical storage systems reveals several complications due to the control of the main equipment involved in the storage process, namely the reactors. The following section gives an overview of the TCES reactors and their limitations.

## 5. TCES Reactors 

A TCHS reactor is a device that contains the storage material and at the same time carries out the process of storing and releasing the energy according to the adopted configuration. Thus, it appears as a crucial component of heat storage processes, and its optimization would allow obtaining very high efficiency of energy storage and restitution. However, few studies have been carried out to develop thermochemical reactors from laboratory prototypes to large-scale project pilots. Those studied and developed so far have a crucial issue of very limited heat and mass transfer in the studied systems [40]. To ensure a sufficient heat and mass exchange inside the reactor, corners where there is no contact between sorbent and sorbate should be avoided or minimized. Indeed, the material selected for storing heat has generally a very high energy density, but the global energy density of the entire system after the storage process remains much lower (generally the third at the end) than that of the materials used [135], which is generally due to the significant heat losses when storing energy. Particular emphasis on all aspects of the reactor (its sizing and optimization) could maximize the storage density of the system, limit energy losses, and thus increase the power of the reactor. TCHS reactor technology consists of two stages, whose mastery could significantly contribute to increasing the maturity of the technology (TRL). These are the microscopic and macroscopic aspects of the design. Most of the published works have focused only on the macroscopic aspect and therefore the published results are still below expectations in the field. However, recently, an increasing number of studies have been focusing on the microscopic aspect after realizing the inevitable importance of this aspect in the technology of reactors.

### 5.1. TCES Reactor Sizing Analysis Criteria and Assumptions

Several authors have worked on the modeling of physical phenomena in thermochemical energy storage reactors [136]. The simulation model used in this paper describes the decomposition of the material inside the reactor. The coupling of heat and mass transfer as well as kinetic equations has been implemented in the Comsol Multiphysics software using the infinite method, to analyze the physical phenomena involved in the reactor. A synthesis of those equations has been made in the following Table 5 to have a general dashboard and a schematic algorithm (Figure 5) for reactor sizing.

Depending on the specificities of the system under consideration, the analysis of experimental data to validate the mathematical models, and the information available on the unknowns of the equations, simplifying hypotheses are formulated to lighten the numerical resolution of these equations

An important aspect in the design of reactors is the design of the heat exchanger that must supply and remove heat before and after the reactions. This aspect also deserves special attention because the efficiency of the system depends on it as well. In addition, in the literature, many thermochemical heat storage projects compact and cylindrical heat exchangers are recommended or used because of the large exchange surface they offer. It should be noted that efficient heat exchanger performance requires small temperature differences throughout the heat exchanger to increase the heat transfer coefficient and thus increase the quality of heat exchange. 

For specific applications, the heat exchangers involved must be dimensioned before any design. In many cases, the sizing is based on the log mean temperature difference. Thus, based on the application requirements, many criteria are integrated into the choice of the selected model, but the thermal power seems to be the most important in thermochemical heat storage, although, for a definitive choice, other parameters such as the pressure drop, the thermal mass, the encumbrance, or the clogging had to be considered. Heat exchanger sizing software such as COMSOL Multiphysics offers a high range of functionality for optimal sizing of the heat exchangers. Moreover, many recent works in thermochemical heat storage have used this software with very satisfactory results [137,138]. The innovative aspect of heat exchangers used in thermochemical heat storage is that most of them are heat exchangers involving a solid and a fluid. The tube containing the heat transfer fluid is in direct contact with the reactive bed, and the heat it contains is directly supplied mainly by conduction to the reactive bed during the loading phase.

### 5.2. TCES Reactor Classification

In TCHS systems, the source to be stored can generally emanate from a solar source that is freely available, from industrial energy waste, and other sources converted into thermal energy. In the literature, there are different approaches to classify TCHS reactors according to their main purpose, but most commonly there is a classification based on the type of system involved, usually direct or indirect [139]. From this classification is derived a categorization according to the operating mode (open or closed). However, more recent classifications based on the components of the system (integrated reactors, separate reactors) or the reaction bed (fixed bed reactors, fluidized bed reactors) are suggested to better design the reactor to be used in the whole storage system. The combination of classifications is used in some literature [117]. Table 6 shows different classifications of reactors, their description, illustrations, the references of the prototypes and the projects developed as well as the efficiency of the reactors obtained with the efficiency of the whole system in which the reactors are installed. 

Additionally, the main differences between TCHS reactors working with a closed system and an open system are classified in Table 7 based on the process involved and the quality of the heat transfer.

Significant results have been obtained in the field of TCHS reactors already offering the existence and feasibility of TCHS prototypes. Most of these projects and work are included in the work and projects on thermochemical heat storage processes, which will be developed in the following section. However, the control of this field remains problematic because it is subject to several limitations that can be highlighted as: The very low heat and mass transfer efficiency of the system.The non-uniform distribution of the energy to be stored within the material reservoir.The uncontrolled corrosion of internal components.The high cost of the conception, which reduces the number of available prototypes.The need for additional gas for fluidization in the case of fluidized reactors.The difficulty of maintaining reactor components.The very high-pressure drops of the transfer fluid at the outlet.The limited number of available simulations works that do not allow a thorough understanding of the physical phenomena involved.

## 6. TCHS Systems Prototypes and Projects

Significant advances have been made not only in the material investigation but also in demonstrative prototypes for TCHS technologies [110,167]. To further identify the problems and limits encountered in the development of TCHS, several synthesis works and reports were published, reviewing the market state, the progress of the studies, and mentioning the future challenges to be tackled [4,128,168]. Relevant experimental results showed the feasibility and great potential, but in the meantime, they unveiled the problems and barriers that researchers need to overcome before going further. In the following section, descriptions of some relevant projects classified by different reaction processes are given in Table 8, and a critical analysis is provided.

It should be noted that one of the important aspects of thermochemical heat storage as mentioned above is the reactor and its optimization so that the energy density of the materials used may be efficiently exploited. Consequently, the recent investigations have been focused for the most part on the characterization of the reactor. The projects listed in Table 8 are very few of the projects in thermochemical heat storage, but the results obtained are very encouraging and deserve to be deepened and improved. More details on the projects carried out at the material and reactor level for TCHS can be found in the recent work carried out by Lin et al. [174].

Three aspects are to be retained from these projects. These are numerical analysis followed by simulation of the reactor; adequate dimensioning of the reaction bed and the experimental work necessary to reduce the margin of error between the numerical simulation and the existing reality to develop an efficient and controllable standard model for future studies; and finally, the development of a large-scale prototype for realistic use of the system. From the analysis of these projects, two reactors stand out: the packed bed or fixed bed reactor and the fluidized bed reactor.

Fixed bed or packed bed reactors have been a significant part of recent projects. In general, a fixed bed consists of a hollow pipe or other container filled with packaged material. The packaging may be arbitrarily filled with small pieces of material, or it may be a specially designed packaging structure, resulting in a system of solid particles in contact and impregnated with a fluid (gas or liquid). A reactor with a fixed bed allows high reaction conversion per unit of mass of material, enables continuous operation, and offers a low-cost operation. Moreover, this type of reactor is highly efficient with a low-pressure drop, as shown by the progress made in some of the projects mentioned above. It is simpler to design and offer a large heat transfer surface that can be contained in a small volume.

A fluidized bed reactor is a type of reactor often used to perform various types of multi-phase chemical reactions. It, therefore, has a considerable advantage for irreversible reactions. Due to its operating mode, it has an idealist contact behavior between the gas and the solid, allowing it to reach a very high energy density and high conversion rates. In this reactor, the material flows by gravity along with several vertical plate heat exchangers that ensure a high heat transfer. Durability studies have shown that this reactor allows stable long-term functioning. 

Although this reactor has important advantages in energy storage, many limitations remain to be overcome. These include the need to provide additional power to ensure a continuous flow, thus leading to additional energy consumption and particle attrition. Furthermore, for most chemical reactions, it is impossible to achieve 100% completion or conversion. Due to its gravitational functioning, there is material erosion and the risk of material sticking to the suspension screw. In addition, special and additional measures are required for steam transport, and there is a certain risk of the material sticking to the heat exchanger surfaces.

In most projects, water is used as the preferred adsorbate because it offers many valuable advantages for this application. Apart from the fact that the transport of water vapor does not require any electrical energy for the transformation cycles, it is a non-corrosive and chemically inert component, it is non-flammable, and it offers a high specific enthalpy of vaporization with high polarity, making it easy to use in sorption systems. In addition, water is a continuous element without rigidity, a chemically stable compound that flows easily and is economically accessible. Water has a high wetting power, which gives it particularly important capillary properties, which is a considerable advantage in the sorption process.

However, due to the low vapor pressure of water, desorption and mass transfer can limit especially low-pressure adsorption processes.

Apart from these types of reactors, increasingly, customized reactor structures are being studied in some recent literature through numerical simulations, even on a laboratory scale, always to find a more efficient solution to the problem of mass and heat transfer, which remains the main obstacle of this technology. 

The problem of mass and heat transfer has remained until now one of the principal barriers to thermochemical heat storage. To alleviate this problem, several techniques are used to get closer to a transitional solution or a compromise between several parameters. D. Aydin et al. [37] carried out a literature review in which they tried to list the methods of improving the heat and mass transfer addressed by some projects. These methods consist essentially of the use of composite materials, which consists of impregnating salt hydrates or other storage materials in porous materials by using different approaches. This technique enhances the rate of sorption of the storage materials that are trapped in the porous matrices, thus increasing the rate of the reaction and improving the mass transfer. Some methods involve the addition of fins to the reactive bed to increase the thermal conductivity of the reactive bed and thus increase the heat transfer. In the same direction, some methods use specific heat exchanger structures to increase the contact surface between the heat transfer fluid and the reactive bed, and some others use compacted reactive bed structures in direct contact with the heat exchanger in a specific structure. 

Apart from this aspect of improving the quality of heat and mass transfer, efforts must be made to ensure that the heat is efficiently stored and used. N’Tsoukpoe et al. [175] have demonstrated that for thermochemical storage in buildings, during the charging phase, about two-thirds of the heat charged into the salt hydrates is lost as condensation heat, which is released into the environment. By using a cascade algorithm in the process of thermochemical heat storage, they have been able to improve the energy and exergy efficiencies of the process by establishing that the heat recovered during the charging period with the use of cascade is about 1.8 times higher than that of the same process without cascade. The same authors in a recent study on the review of long-term thermochemical heat storage systems for residential applications have shown that the volumetric densities of energy storage displayed by processes based on solid hydrates are prohibitive for the long-term heat storage applications [176]. As a result, an overestimation of the data is often observed. It would be more interesting if the simulation works exploited the parameters and data from their experimental studies with a certain level of precision.

As far as the numerical simulation is concerned, the development of recent advanced simulation tools such as COMSOL Multiphysics, Ansys fluent, and Dymola Modelica has made it possible to perform the virtual functioning of the storage systems, mainly the reactors, to visualize the behavior of the latter under different modes and operating conditions over a given time. This allows exploring the possible configuration and combination options for system optimization to avoid the financing of unpromising prototype projects and to save time. Thus, through numerical simulation, it is possible to observe the heat and mass transfer phenomena in 3D within the reactor. Additionally, it allows the coupling of physics, mainly chemical reaction physics, with heat transfer, which represents a major asset in the dimensioning of reactors. Certain parameters such as the reaction rate, the permeability of the reactive bed, the energy density of the system, etc., can therefore be visualized and optimized.

However, despite the immense importance of numerical simulations and the advantages they offer, the field is still under-exploited and has several limitations for further improvement. Indeed, before implementing the numerical model in the simulation software for the resolution of the equations governing the different physics involved, several simplifying hypotheses are taken into consideration to reduce the complexity of the model. Several simplifying errors are made, and in some cases, the simulation results are quite different from those obtained after the experimental work, and in other cases, they are close but with a certain margin of error. Additionally, it should be noted that for the numerical model, some parameters used are determined by experiments that make simulation data depending on the experimental results because the accuracy of the experimental data would also affect the accuracy of the results after simulation. Furthermore, some very promising results after the simulation work could not be implemented in real prototype systems due to the complexity of the optimization methods used, the cost of the technology involved, and the complexity of the simulation data and algorithms. 

Although it is the most promising of the TCHS processes with very high heat densities and temperature ranges, this storage method suffers from a lack of investigations and projects, especially in the field of heat and mass transfer, which reveal the main limitations of this process. Additionally, the complexity of the technologies used makes the systems very expensive and complex to maintain. Future works and approaches in this area can reflect on the approaches, combining with other sciences, namely computational fluid dynamics, heat, and mass transfer in solid and in porous media, to fill the gaps in this process. Furthermore, the optimization algorithms used in the simulation works should be clear enough to allow the implementation of the results in the real prototype scale.

## 7. Conclusions and Prospects

In the present paper, the features of TCHS are discussed. The research status and the purposes of the available systems are reviewed, and their advantages, as well as their actual drawbacks, are analyzed. Thus, TCHS can be summarized into four main branches namely: TCHS processes, TCHS materials, TCHS reactors, and TCHS system devices. The main conclusions are as follows:TCHS is a wide area of investigation more than as discussed in one paper.Absorption and adsorption processes are commonly used for space heating purposes and applications that require a low or middle grade of temperature, whereas the chemical reaction process is used for high energy density and high temperature.Magnesium chloride has received considerable interest in recent work with the increasing use of open-circuit reactors for building heating applications and inorganic hydroxide material for high-temperature applications.The shape of the reactors, as well as the correct choice of the reactive bed, appeared to be very important. It should therefore be emphasized that before any experimental work and any prototype design, a numerical simulation through the above-mentioned software must be performed, and the simulation works should be clear enough and realistic to allow the implementation in a prototype device for experimentationsThe sizing of heat exchangers is an integral part of reactor sizing, and their efficiency has an impact on the reactor performance.Before any design of the system, it is necessary to take into account the real application required (building heating, industrial hot temperature process, etc.).Closed TCHS reactors require a heat exchanger to provide or remove the heat of the reaction. It involves a lot of technical components but allows better control of the reactor and offers better reaction kinetics.The open TCHS reactor, because it operates at atmospheric pressure, overcomes these constraints, offering a more simplified and less economical design, but it cannot provide better control of the reactor. New technology must therefore be found to combine these two operating systems. We suggest the type of open system capable of operating with a heat exchanger connected to an external open system.For both TCHS reactors, many barriers remain to be overcome. Particularly on the geometries of the devices, which do not ensure an optimal operation. Heat exchanger selection criteria, the control of the transfer phenomena within the reactors, and the problems of recycling materials after the reactions are some. While the cycles and lifetimes of the materials altered after the reactions, the problem of corrosion within the reactors as well as the high thermal losses deteriorates the performance of the equipment. The problems of adapting specific devices for each type of application involved, and the difficulties of maintenance of the systems due to their complexity complete the barrier list.Numerical simulations are of crucial importance in the dimensioning of thermochemical heat storage systems. In this study, a dashboard has been prosed for this purpose.For the visualization of physical phenomena occurring within the reactor, more recently the COMSOL Multiphysics software is increasingly being used since it offers a 3D model for the resolution of equations with a very satisfactory mesh system and calculation accuracy. The Trnsys Simulation software is also used in the case of a macro-scale simulation involving the production or storage of energy until its end-use. It should therefore be emphasized that before any experimental work and any prototype design, a numerical simulation through microscale and macroscale system software must be performed.Future research directions must take into account:The problems that hinder TCHS technology separately: microscopic aspect and macroscopic aspect.The enhancement of the thermal conductivity of the storage materials and the stability of the reaction cycles.The development of several models of the reactive bed, allowing better storage and a total reaction rate of the materials.The investigation of coupling between the physics of heat transfer in porous media, chemical reactions, and the transport of diluted species in solution.The analysis of corrosion problems inside reactors.The numerical simulation of a varied range of a combination of reactor and storage materials, to have a consistent database for technological choices.Intensifying of experimentation efforts to validate and promote numerical models and simulations works, and application of TCHS technology through technical and economic feasibility analysis.

## Figures and Tables

**Figure 1 entropy-23-00953-f001:**
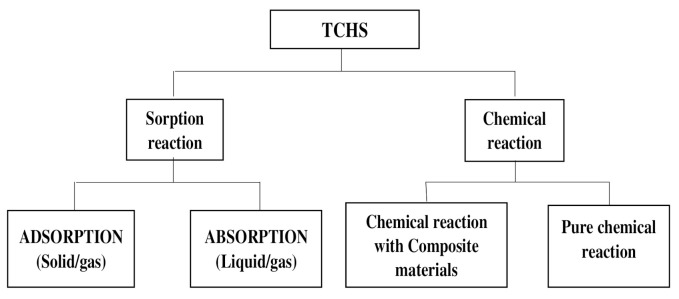
Classification of TCHS processes.

**Figure 2 entropy-23-00953-f002:**
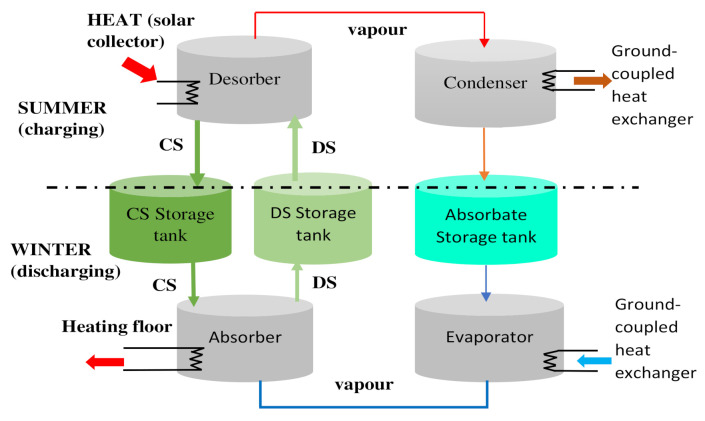
Absorption storage system (DS: diluted solution; CS: concentrated solution).

**Figure 3 entropy-23-00953-f003:**
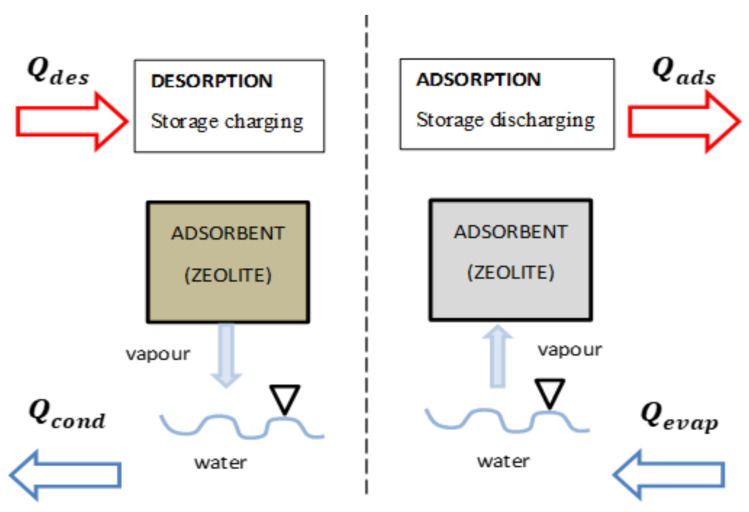
Working principle of adsorption thermal energy storage.

**Figure 4 entropy-23-00953-f004:**
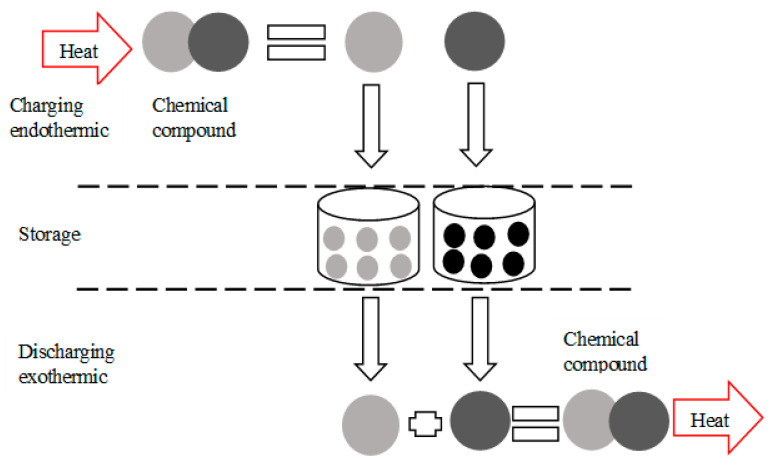
Working principle of chemical thermal energy storage.

**Figure 5 entropy-23-00953-f005:**
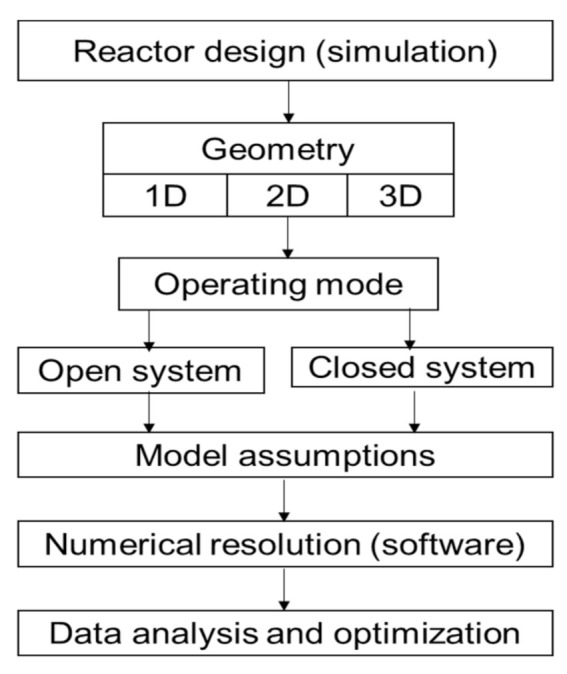
Schematic description of the microscopic analysis of the TCHS system.

**Figure 6 entropy-23-00953-f006:**
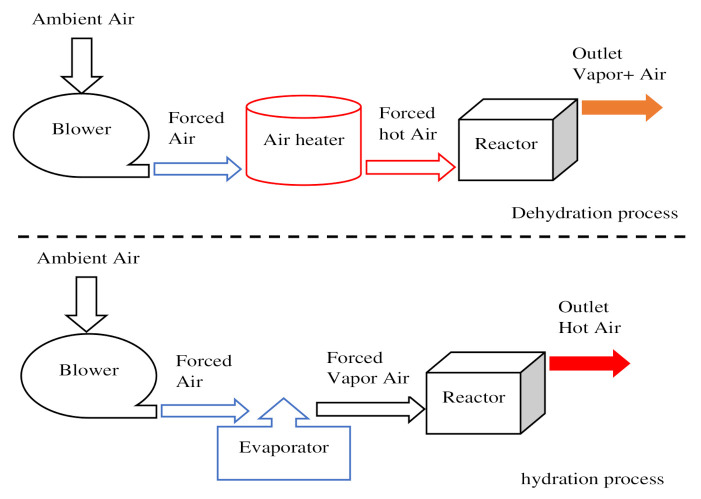
Main steps for the reversible reaction in an open TCHS system.

**Figure 7 entropy-23-00953-f007:**
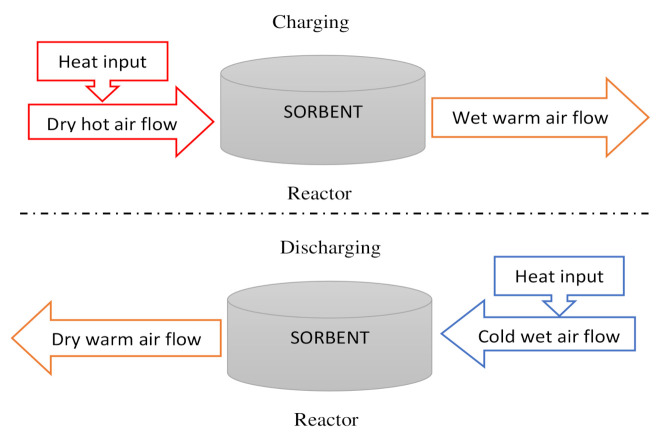
Simplified diagram of the main steps for the reversible reaction in an open TCHS system.

**Figure 8 entropy-23-00953-f008:**
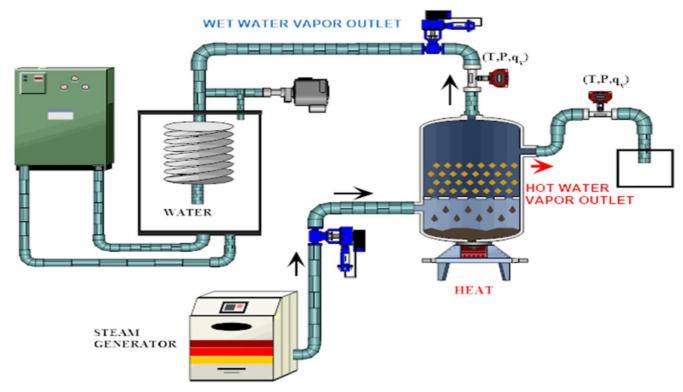
Closed TCHS system.

**Figure 9 entropy-23-00953-f009:**
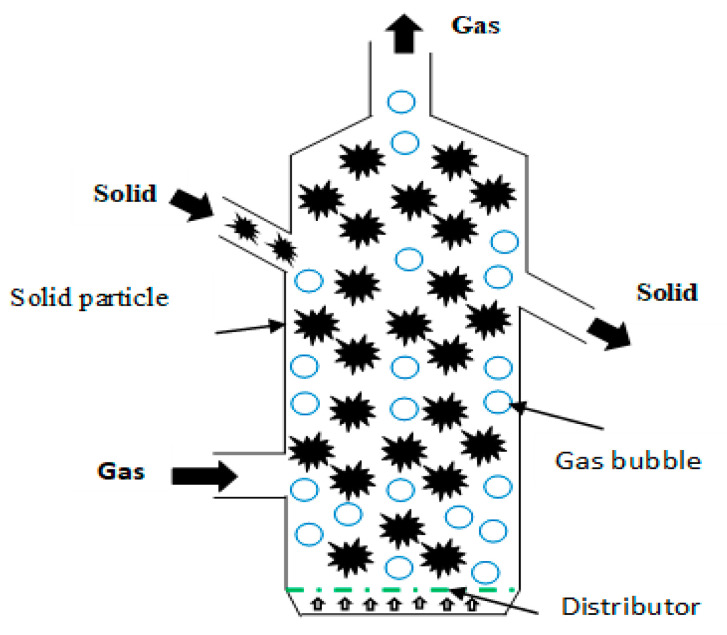
Fluidized bed reactor 2D view.

**Table 1 entropy-23-00953-t001:** Summary of recent review articles on low and medium temperature TCES systems.

Author(s)	Highlights	Refs
N’Tsoukpoe et al., (2009); Tatsidjodoung et al., (2013)	Investigated the materials used for TCHS processes for household applications. Outlined and summarized projects involved in long-term sorption heat storage up to 2009.	[5,6]
Cot-Gores et al., (2012)	Examined reactions based on TCHS. Emphasized microscale to large-scale prototypes of solid–gas systems based on sorption reaction.	[7]
Yu. et al., (2013)	Reviewed the TCHS based on sorption technologies. Described the storage mechanisms and the studies carried out with open and closed circuits.	[8]
Sole et al., (2015); Fopah-Lele and Tamba (2017)	Featured various reactor designs for gas-solids and chemical TCHS systems, designed from bench to pilot scale. Highlighted different storage materials characterization and modeling techniques (2D and 3D views of the physics).	[9,10]
Krese et al., (2018)	Stressed the importance of solar energy integration in TCHS systems for household applications. Suggested numerical modeling of open and closed sorption heat storage systems.	[11]
Lizana et al., (2018)	Reviewed recent advances and prospects for heat storage systems in the building. Examined several materials and mechanisms for thermal heat storage systems.	[12]
Kuznik et al., (2018)	Clarified and presented the principle of physisorption heat storage for building applications. Targeted different storage materials with the corresponding reactors for prototyping and experimental studies.	[13]
Sunku Prasad et al., (2019)	Examined TCHS systems functioning at high temperatures. Presented materials as well as their reaction mechanisms suitable for high-temperature heat storage. Outlined the available reactor models suitable for solid–gas reaction materials. Discussed cyclic analysis of storage materials and prospects for future research.	[3]
Desai et al., (2020)	Emphasized storage materials for low- and medium-temperature heat storage. Provided a comprehensive overview, description, and analysis of recent research studies undertaken at both bench and pilot scales. Suggested storage system configurations for solar cooling and heating applications.	[4]

**Table 2 entropy-23-00953-t002:** Characteristics of some sorbents.

Adsorption							
Material	Operating Temperatures	Energy Density of the Bed	Prototype Energy Density	Storage Capacity kWh	Discharge Power kW	Discharge Time h	References
Mesoporous silicates	Charge: 88 °C Discharge: 42 °C	119	52.3	27.4 (808 kg of anhydrous silica gel).	2.87	9.5	[118]
	Charge: 88 °C Discharge: 42 °C	50	33.3	13	0.5–1	13–26	[6,121]
	Charge: 180 °C Discharge: 30 °C	180	57.8	1	0.8–1.8	0.56–1.25	[6,122]
Zeolite 4A–H_2_O	Charge: 180 °C Discharge: 35 °C	160	120	12 kWh (70 kg of anhydrous 4A zeolite)	1–1.5	8–12	[6]
Zeolite 13X–H_2_O	Charge: 130 °C Discharge: 65 °C	124	NA	1300	135	9.6	[121,123]
Zeolite H_2_O	Charge: 135 °C Discharge:140 °C	NA	NA	2400 (14 t of dehydrated zeolite)	NA	10	[124]
Zeolite H_2_O	Charge: 180 °C Discharge: 60–50 °C	140–220	NA	NA	134	NA	[29,118,125]
Activated carbon Methanol	Charge: 95 °C Discharge: 35 °C	60 (Simulated, bed energy density)	NA	NA	NA	NA	[126,127]
Absorption							
NaOH/H_2_O	Charge:100–150 °C Discharge: 40–65 °C	250	5	8.9	1	8.9	[41]
LiCl/H_2_O	Charge: 46–87 °C Discharge: 30 °C	253	85	35	8	4.4	[84]
CaCl_2_/H_2_O	Charge: 70–80 °C Discharge: 21 °C	NA	116 (data from simulations)	15	0.03 −0.560	27–500	[121]
LiBr/H_2_O	Charge: 75–90 °C Discharge: 30–38 °C	251	NA	8	1	8	[89,128]
CaCl_2_/H_2_O	Charge: 95 °C Discharge: 35 °C	NA	200 (Simulation, prototype energy density)	NA	NA	NA	[8]

NA: not available.

**Table 3 entropy-23-00953-t003:** The characteristics of material couples suitable for heat storage in residential applications.

Reaction Equations	Operating Conditions °C	Heat Storage Density	Characterization Level
MgSO_4_·7H_2_O → MgSO_4_ + 7H_2_O	Charge: 122–150 Discharge: 122	1512 of MgSO_4_ (theoretical)	Material scale ECN project: Characterization, experimental tests, a sample of 10 mg
MgCl_2_·6H_2_O → MgCl_2_·2H_2_O + 4H_2_O	Charge: 115–130 Discharge: 35	2170.8 of MgCl_2_·2H_2_O	Material scale IEC Project: Material characterization. Sample of 250 mg. Stabilization with zeolite 4A is to be further considered.
MgCl_2_·6H_2_O → MgCl_2_·H_2_O + 5H_2_O	Charge: 150 Discharge: 30–50	-	Material scale ECN project: Material characterization, a sample of 300 g of material.
CuSO_4_·5H_2_O → CuSO_4_·H_2_O + 5H_2_O	Discharge: 40–60 (heat supply at T ≥ 40 to ignite discharge)	2066.4 of CuSO_4_·H_2_O (theoretical)	ITW project: Material characterization, a sample of 100 mg
CaCl_2_·2.3H_2_O → CaCl_2_ + 2.3H_2_O	Charge: 150 Discharge (temperature lift): T = 6.2 (reactor and evaporator both at 25) T = 10 (reactor at 50 and evaporator at 10)	-	ECN project: Material characterization, a sample of 40 g
Bentonite + CaCl_2_	Discharge: 35	667 of composite material	ITW Project: Material characterization
Kal(SO_4_)_2_·12H_2_O → Kal(SO_4_)_2_·3H_2_O + 3H_2_O	Charge: 65 Discharge: 25	864 of Kal(SO_4_)_2_·3H_2_O	Reactor scale: PROMES CEA-INES Project, 25 kg of Kal(SO_4_)_2_·12H_2_O
Al_2_(SO_4_)_3_·18H_2_O → Al_2_(SO_4_)_3_·5H_2_O + 13H_2_O	Charge: 150 Discharge (temperature lift): T = 9.8 (reactor and evaporator both at 25) T = 10 (reactor at 50 and evaporator at 10)	-	Reactor scale. ECN Project: Sample of 40 g
Na_2_S·5H_2_O → Na_2_S·1.5H_2_O + 4.5H_2_O	Charge: 83 Discharge: 35	2808 of Na_2_S·1.5H_2_O	Reactor scale, ECN project: SWEAT prototype, 3 kg of material.
SrBr_2_·6H_2_O → SrBr_2_·H_2_O + 5H_2_O	Charge: 70–80 Discharge: 35	216 of SrBr_2_·H_2_O	Reactor scale, PROMES CEA- INES Project: SOLUX
CaCl_2_·2H_2_O → CaCl_2_·H_2_O + H_2_O	Charge: 95 Discharge: 35	720 of CaCl_2_·H_2_O	Reactor scale, BEMS: a theoretical study

(–): not available.

**Table 4 entropy-23-00953-t004:** Chemical storage materials.

Reaction Equations	Reaction Temperature, °C	Energy Storage Density	Advantages	Drawbacks
Ammonia decomposition material ${\mathrm{NH}}_{3}⇋\frac{1}{2}N_{2}+\frac{3}{2}H_{2}O$	400–700	67 kJ·mol^−1^	- The decomposition/synthesis reaction has good reversibility, and there is no side reaction. - Working mediums are fluids that are easy to transport. - Properties of NH_3_, N_2_, and H_2_ are stable in the operating temperature range, and the sources of reactants and products are abundant and cheap. - Under normal temperature and medium pressure, the density difference between NH_3_, N_2_, and H_2_ is large. Therefore, they can be separated automatically when stored in the same storage tank. - Supported by extensive industrial experience in ammonia synthesis and treatment, operational processes and design guidelines can be referred to the existing specifications.	- Economic viability of the system is closely related to its initial capital cost, in particular, the cost of the reactor wall material. - The harsh chemical reaction conditions. - The need for high-pressure operation. - The effect of high-efficiency catalyst - Operating cost of the system is quite high.
Inorganic hydroxide material $\mathrm{Ca}{\left(\mathrm{OH}\right)}_{2}⇋\mathrm{CaO}+H_{2}O$	350–900	300 kWh·m^−3^	- Nontoxic and cheap. - High-temperature Ca(OH)_2_ TCES systems. - High energy storage density. - The reactants are safe, economical, and environmentally friendly. - Capacity of storing the waste heat or excess - Heat in the cogeneration process for a long time without heat loss. - The Ca(OH)_2_/CaO energy storage system has a rapid and effective reaction kinetics. - High reaction enthalpy. - The reaction does not require a catalyst and is insensitive to pressure requirements. - The reaction is simple and easy to obtain; for a long storage life, there is no energy consumption and few components in operation.	- Further technology development. - The heat transfer performance of MgO and Mg(OH)_2_ is poor. - The cycling stability is poor (sintering of MgO products result in grain growth and loss of pore volume, which reduced the rehydration kinetics of MgO). - Thermal conductivity of Ca(OH)_2_/CaO system is poor.
Inorganic hydroxide material $\mathrm{Mg}{\left(\mathrm{OH}\right)}_{2}⇋\mathrm{MgO}+H_{2}O$	100–167	380 kWh·m^−3^		
Methane reforming material ${\mathrm{CH}}_{4}+{\mathrm{CO}}_{2}⇋2\mathrm{CO}+2H_{2}$	700–860	247 kWh·m^−3^	- High temperature. - High energy storage density.	- Operating cost of the system is quite high.
Methane reforming material ${\mathrm{CH}}_{4}+H_{2}O⇋\mathrm{CO}+3H_{2}$	600–950	250 kWh·m^−3^		
Carbonate decomposition material ${\mathrm{CaCO}}_{3}⇋\mathrm{CaO}+{\mathrm{CO}}_{2}$	700–1000	692 kWh·m^−3^	- Innocuous, environmentally friendly, cheap, and easily available characteristics. - High energy storage density. - Carbonation of CaO is a promising technique for capturing CO_2_ from flue gas.	- Poor cycle stability of CaO. - Most natural calcium-based materials exhibit a gradual deactivation.
Metal hydride material ${\mathrm{MgH}}_{2}⇋\mathrm{Mg}+H_{2}$	250–500	75 kJ·mol^−1^	- Cyclic stability of hydrogen storage materials.	- Poor thermal conductivity and reaction kinetics performance - More expensive engineering designs for efficient heat transfer - Hydrogen storage, tank manufacturing, and safe operation of the system are large challenges - Cost of the alloy is high
Redox material $2{\mathrm{BaO}}_{2}⇋2\mathrm{BaO}+O_{2}$	127–1027	77 kJ·mol^−1^	- Reversible redox reaction - Air can be used either as heat transfer fluid or as a reactant in direct contact with storage materials (metal oxides) without additional heat exchangers. - Heat loss can be greatly reduced - High operating temperature.	- Poor cycling stability. - Complete transformation is limited by mass transfer and crust on the material surface. - Toxic and expensive. - Challenges in terms of sintering, softening, and agglomeration.
Redox material $2{\mathrm{Co}}_{3}O_{4}⇋6\mathrm{CoO}+O_{2}$	700–850	205 kJ·mol^−1^		

**Table 5 entropy-23-00953-t005:** Dashboard for TCHS reactor simulation.

Dashboard 1: Open System	Dashboard 2: Closed System
Kinetic equation $\begin{array}{c} \frac{\partial \alpha }{\partial t}=\overset{f}{\underset{i=1}{\sum }}A_{i}\cdot \exp\left(\frac{-E_{i}}{\mathrm{RT}}\right)\;\cdot f_{i}\left(\alpha_{i}\right)\cdot \left(1-\frac{P_{i}}{P_{\mathrm{eq}}}\right) \end{array}$ Mass equation $\left(1-\varepsilon_{s}\right)\frac{\partial \rho_{s}}{\partial t}=\rho_{s}\frac{\partial \alpha }{\partial t}\left(\mathrm{solid}\right)$ $\varepsilon_{s}\frac{\partial \rho_{v}}{\partial t}=S_{w}-\nabla \left(\rho_{v}\vec{u}\right)+D_{g}\Delta \rho_{v}\;\left(\mathrm{gas}\right)$ $\frac{\partial }{\partial t}\left(\varepsilon_{s}\rho_{\mathrm{air}}\right)+\nabla \left(\rho_{\mathrm{air}}\vec{u}\right)=S_{w}\;\left(\mathrm{moist}\;\mathrm{air}\;\mathrm{mixture}\right)$ Energy equation $\left(\rho_{s}C_{p,s}+\varepsilon_{s}\rho_{\mathrm{air}}C_{p,\mathrm{air}}\right)\frac{\partial T}{\partial t}=\nabla (\lambda_{\mathrm{eff}}\cdot \nabla T)-\rho_{\mathrm{air}}C_{p,\mathrm{air}}\vec{u}\nabla T-S_{w}\frac{\Delta h_{r}^{0}}{v}$	Kinetic equation $\begin{array}{c} \frac{\partial \alpha }{\partial t}=\overset{f}{\underset{i=1}{\sum }}A_{i}\cdot \exp\left(\frac{-E_{i}}{\mathrm{RT}}\right)\;\cdot f_{i}\left(\alpha_{i}\right)\cdot \left(1-\frac{P_{i}}{P_{\mathrm{eq}}}\right) \end{array}$ Mass equation $\left(1-\varepsilon_{s}\right)\frac{\partial \rho_{s}}{\partial t}=\rho_{s}\frac{\partial \alpha }{\partial t}$ $\varepsilon_{s}\frac{\partial \rho_{v}}{\partial t}=vS_{w}-\nabla \left(\rho_{v}\vec{u}\right)+D_{g}\Delta \rho_{v}$ $\alpha =\frac{m_{0}-m}{m_{0}-m_{f}}\left(\mathrm{experiment}\right)$ $\alpha =\frac{m_{0}-m}{m_{0}-m_{f}}\left(\mathrm{simulation}\right)$ Energy equation $C_{p,s}\left(1-\varepsilon_{s}\right)\rho_{s}\frac{\partial T}{\partial t}=\nabla (\lambda_{\mathrm{eff}}\cdot \nabla T)-\rho_{v}C_{p,v}\vec{u}\nabla T+\frac{\rho_{s}}{M_{s}}\frac{\partial \alpha }{\partial t}\Delta h_{r}^{0}$

**Table 6 entropy-23-00953-t006:** Reactor classifications.

Classification	Reactors	Subclassification	Reactor Description	Reactor Efficiency	Whole System Efficiency	Prototypes References
System involved	Indirect reactor	-	The material reservoir is directly heated to either heat the material or the heat transfer fluid by conduction.	21—41%	10–20%	[140,141,142,143]
	Direct reactor	Open reactor	Reactants are heated by heat source input through an opened receiver aperture. Moist air atmospheric acts as a mass and heat carrier fluid. Use of gas diffusers to supply or collect the moist air (Figure 6 and Figure 7). Use of external heat exchanger to carry out the heat.	25–50%	15–35%	[144,145]
		Closed reactor	Reactants are heated by heat source input through a closed receiver aperture and the system is isolated from the atmospheric environment. Use of a gas diffuser and an internal heat exchanger to collect or supply the heat of the reaction. An evaporator is required to generate steam for the hydration phase and a heat source (Figure 8)	40–64%	15–40%	[146,147,148,149]
Reactors type	Stacked bed reactor	Fixed bed reactor Mobile bed reactor Rotary bed reactor	The material is packed inside the reactor heated by air/HTF flowing through the material bed/heat exchanger. The heat exchanger is used to carry out the heat of the reaction, and the material is replaced after full conversion is achieved. The HTF can flow cocurrent upflow/downflow or countercurrent.	12–69%	12–42%	[26,127,130,145,150]
	Fluidized bed reactor	Vibrated bed reactor Blown bed reactor	The material is fed in the reactor and the fluid is fed through granular solid material. The high velocity of the fluid generates a suspension of material particles that act as if they were a fluid, which increases the quality of the heat transfer (Figure 9).	15–75%	20–60%	
System components	Separate reactor		Dissociation between the thermal power and the storage capacity. There is no need for a steam or heat exchanger integrated into the reactor. Only the required amount of reactant is heated.	28–75%	20–65%	[26,151,152,153,154,155,156,157,158,159,160,161,162,163]
	Integrated reactor	Two phases reactor	Contains the material and the sorption pair reactant (air or steam). Need of heat exchanger to carry out the heat of the reaction.	14–65%	15–55%	[127,164,165,166]
		Three phases reactor	Contains the material and the sorption pair reactant (air or steam). The addition of a second working fluid is disconnected from the other and acts as a heat exchanger. Addition of material transport system.	14–85%	15–65%	

**Table 7 entropy-23-00953-t007:** The differences between TCHS reactors working with a closed and open system.

Operating Circuit	Operating Pressure	Heat and Mass Transfers and Heat Storage	Design and Dimensioning
Closed system	The necessity of an evaporator to produce steam during hydration reaction	- Need for a heat exchanger to provide or remove the heat of reaction. - Heat exchange is the principal obstacle to the reaction.	Strong technological constraints and manufacturing for the reactor and the evapo/condenser design.
Open system	The steam is provided by moist air coming from the environment	- HTF and reactive gas are combined in a single flow. - System configuration without internal heat exchanger. - Mass exchange remains the main factor limiting the storage mechanisms. - Permeability after completion of the reaction is a decisive criterion for dimensioning the reaction bed. - System operating with humid air and suitable for long-term heat storage for household applications. - Higher thermal power with the absence of evaporator and volume exchanger.	- Allows simpler manufacturing of the reactor. - Easier conception and management - Lower cost. - Necessary use of a blower to drive humid air across the reactor.

**Table 8 entropy-23-00953-t008:** TCHS systems prototypes and projects.

Investigation (Authors) & Years	Nature and Purpose	Storage Process	Progress & Contribution	Storage Materials	Reactor Type	Storage Density/ Storage Temperature	Descriptions	Refs.
Zhang et al., (2014) Shanghai Jiao Tong University (2020)	Space heating and domestic hot water Lab-scale	Absorption	Single-stage absorption; Multiple functions using: production of child water at 7 °C, heating water at 43 °C for space heating and domestic hot water at 65 °C	$\mathrm{LiBr}/H_{2}O$	Integrated	42, 88, and 110 $\mathrm{kWh}\cdot m^{-3},$ respectively for child, heating, and hot water	Single or multiple storage tanks (Two or more) can be integrated with the absorption chiller/heat pump. Both the refrigerant and solution might be stored simultaneously. The refrigerant storage is in association with a condenser. The weak solution storage is in association with an absorber, and the strong solution is in association with the generator.	[43,169]
French National Research Agency (2017)	Space heating Reactor scale, prototype, commercial	Absorption	Development of low-temperature heat storage system	$\mathrm{KCOOH}/H_{2}O$	Closed	$T_{\max}$ = 60 °C	Four main components: a desorber, an absorber, a condenser, and an evaporator, Two solution storage tanks (for the diluted and concentrated solution) and an absorbate storage tank. The solution can crystalize in the solution storage tank in this process. A sandwich grooved vertical plate configuration was then chosen for the heat exchanger	[50]
EMPA, COMTES (2014)	Space heating Reactor scale, prototype, commercial	Absorption	Development of commercial prototype energy storage system Achievement of an increase in volumetric energy density compared to hot water storage	$\mathrm{NaOH}/H_{2}O$	Closed	250 $\mathrm{kWh}\cdot m^{-3}$	Two chambers are connected Chamber 1 contains the sorbent and functions as desorber during charging and as absorber during discharging. Chamber 2 works as condenser and evaporator, respectively, containing the sorbate.	[170]
Fopah-Lele (2015)	Numerical investigation Space heating	Adsorption	Modeling of the charging and discharging phase of a storage system with low thermal energy. Study of the influence of the performance of parameters such as temperature, pressure, and heat transfer coefficient on the charging process. 3D view of the heat transfer and the behaviors of the reactor during the process.	${\mathrm{SrBr}}_{2}\cdot 6H_{2}O$	Closed	531.77 $\mathrm{kWh}\cdot m^{-3}$	Numerical analysis of two types of heat exchanger: plate-fin and helical coil heat exchangers embedded in a ${\mathrm{SrBr}}_{2}\cdot 6H_{2}O$. A reactor based on a honeycomb heat exchanger concept was design. The model is solved with the COMSOL Multiphysics software. The analytic results were implemented on a lab scale and prototype for validation of the model.	[138]
Q. Ranjha (2017)	Numerical investigation Industrial process	Adsorption	Heat and mass transfer 3D simulation using a novel structure of reactor with $\mathrm{Ca}{\left(\mathrm{OH}\right)}_{2}/\mathrm{CaO}$ powders. Optimization technic for selecting the appropriate structure of the reactor.	$\mathrm{Ca}{\left(\mathrm{OH}\right)}_{2}/\mathrm{CaO}$	Closed	$T_{\max}$ = 550 °C	Indirectly heated fixed reaction beds of circular and rectangular cross-sections; heat transfer fluid could flow in co-current, counter-current, or cross-flow to the reaction gas; COMSOL Multiphysics software is used for the simulation of the reactor.	[137]
COMTES (2012)	Lab-scale prototype Space heating	Adsorption Absorption	Development of liquid and solid sorption systems for seasonal heat storage purposes	Zeolite 13XBF	Closed fixed bed	$T_{\max}$ = 75 °C	Closed modular solid sorption system with an additional backup heater; prototype with a reactor of approximately 300 L and 164 kg.	[117]
Shanghai Jiao Tong University (2017)	Pilot-scale Space heating	Adsorption	Locomotive air conditioning system enhancement	Zeolite 13X	Closed	$T_{\max}$ = 125 °C	Closed sorption system; the heat source is provided by high-temperature gas exhausted from an internal combustion engine; use of one adsorber and a cold storage tank; the cooling effect is transferred to the cabin by chilled water.	[119]
Lauren Farcot et al., (2019)	Building heating	Adsorption	Study of the impact of the air humidity at the reactor inlet on the reactor performances. The feasibility of continuous thermochemical heat storage in a moving bed reactor with hydrated salt.	${\mathrm{SrBr}}_{2}\cdot H_{2}O$/ ${\mathrm{SrBr}}_{2}\cdot 6H_{2}O$	Open moving bed	T = 41 °C	Moving bed reactor with a wall in stainless steel to avoid corrosion. The air diffuser and collector are separated from the reactor by stainless steel. The cross-section of the reactor can be adjusted (reduced) by adding stainless steel walls along with the mesh. A stainless-steel reservoir receives the salt that falls from the rotary valve at the bottom of the reactor. Air can be flown through the reactor at a temperature between 0 and 100 °C.	[38]
Abanades (2018)	Industrial process	Chemical	Design and demonstration of a high temperature solar-heated rotary tube reactor for continuous particle calcination.	${\mathrm{CaCO}}_{3}/\mathrm{CaO}$	Indirect rotary tube	500–1600 °C	The reactor is composed of a cavity-type solar receiver for radiation absorption; external heating by concentrated solar energy is provided; indirect heating of the reactants; the heat is transferred to a rotary tube, and the reactive particles are continuously injected into the rotary tube.	[130]
Zhejiang University of Technology (2018)	Power generation	Chemical	Design of a methanation reactor for producing high-temperature supercritical carbon dioxide in solar thermochemical energy storage	${\mathrm{CH}}_{4}/{\mathrm{CO}}_{2}$	Tubular packed bed	700 °C	The system is composed of an endothermic and exothermic reactor at the inlet and outlet, respectively, each connected to a heat exchanger. Two materials vessels are used. The first to provide the methane to the endothermic reactor for decomposition under solar heating. The second is to store the product of the decomposition, which is pumping in the second reactor for ${\mathrm{CO}}_{2}$ production.	[171]
Y. A. Criado (2014)	Power generation	Chemical	Analyze a thermochemical energy storage process using a hydroxide calcium chemical loop.	$\mathrm{CaO}/\mathrm{Ca}{\left(\mathrm{OH}\right)}_{2}$	Fluidized bed reactor	260 $\mathrm{kWh}\cdot m^{-3}$	Use of a single fluidized bed reactor alternating from hydration to dehydration conditions and two solid storage silos feeding solids continuously to the fluidized bed. Operation at atmospheric pressure is assumed for simplicity.	[164]
Edwards et al., (2012)	solar power plants	Chemical	Developed a calcium looping CSP plant. Determined the operating conditions required to achieve satisfactory operation of the plant.	$\mathrm{CaO}/{\mathrm{CaCO}}_{3}$	Fluidized bed reactor	875 °C	Use of two reactor units operating independently. A solar calciner using concentrated solar energy to start the reaction. A carbonator recombines the product of the reaction thus releasing heat to the turbine.	[172]
General Atomics (2O11)	Power generation	Chemical	Modified a storage cycle to yield elemental sulfur as a by-product, which is then stored and later used as a combustible to generate power.	Sulfur-Based		1200 °C	Use of two distinct turbines. The first one is actuated by the flue gas leaving the combustion chamber. Use of heat exchanger to drive the heat to the second turbine powering a Rankine cycle.	[173]

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
