# Peer review of "Recent Status and Prospects on Thermochemical Heat Storage Processes and Applications"

_entropy, 2021, doi:10.3390/e23080953_

Round 1
Reviewer 1 Report
The authors included all the concerns that I previously comment. It can be considered publication.
Author Response
Dear Reviewer,
Thank you for the valuable suggestions and comments. We have further checked the current version in order to improve the English.
Best,
Reviewer 2 Report
- Please define why in some cases TCES term is used and in other cases TCHS term is used. Several times throughout the text there is mixing of these two terms. Please try to maintain a more consistent use of the terms.
- In page 5, line 91, the authors use the term "new process". Please define what is this "new process".
- In Figure 1 there are typos (chemical "rection"). Please correct.
- Figure 5, 6, 11. Please improve quality.
- Line 308. Please put reference.
- The authors have not included any work on structured reactors for high temperature thermochemical storage from concentrated solar radiation (e.g. based on cobalt oxides), as well as generally on metal oxide two-step thermochemical cycles. Please reconsider.
- There is an issue with the references as they are cited throughout the text. The references as they are cited in the text seem not to follow the correct sequence. Please check and correct the list of references and how citation is made in the text.
- From Ref. [60] line 149 to ref [76] line 178 there are missing reference numbers. Some appear in line 214 ref. [64-87].
- Then in line 220 the references jump to numbers [103]
- Then in line 240 references [92-118] are cited, but then in line 255 it jumps to [126] and in line 258 to [134-136]. etc.
Author Response
Please see the attachment.

This manuscript is a resubmission of an earlier submission. The following is a list of the peer review reports and author responses from that submission.
Round 1
Reviewer 1 Report
The authors reviewed a manuscript entitled "Recent Status and Prospects on Thermochemical energy storage processes and applications". The report will contribute its part in thermal energy storage fields. It can be considered publications after considering the following major points, detailed below:
- The author should address gaps that the present report will fill compare with previously reported review works. In general, what is new in this report compared with other extensive reports.
- Although the review includes enough literature citations, it still lacks very recent literature that is reported currently. I recommend revising and include recently reported related to TCES.
- please avoid unnecessary punctuation marks including (.) from the end of the title.
- In the abstract, the authors claimed "Thermochemical energy storage (TCES) emerges as the most promising method to store thermal energy for further use". How thermal energy storage (TES) can store itself? I think it is the material like phase change materials that can participate to store TES.
- Some figures and illustrations are not clear including Graphical Abstract. please avoid unnecessary figure stretching like Figure 6.
- Surprisingly, the introduction is not supported by a single reference. The introduction is still not enough., need additional information.
- Please thoroughly revise abbreviations, symbols, punctuation, and other required modifications. Meanwhile, please write a similar font style for Figures.
- The conclusion part needs critical revision, please dig out the main points and outlooks.
Author Response
Dear reviewer #1,
we have appreciated your comments in order to improve this paper.
Point 1: The author should address gaps that the present report will fill compare with previously reported review works. In general, what is new in this report compared with other extensive reports.
Response 1: This paper highlights the difficulty of properly addressing the problems encountered in thermochemical heat storage when considering all the disciplines involved in a single paper. It is suggested that thermochemical heat storage be analysed from the microscopic point of view (simulation and analysis of the physics governing the reactions) to the macroscopic point of view (system efficiency and optimization techniques). Besides, the paper suggested a general algorithm and dashboard that can be used for microscopic analysis. The paper recommended a new perspective for future analysis in thermochemical heat storage study.
Point 2: Although the review includes enough literature citations, it still lacks very recent literature that is reported currently. I recommend revising and include recently reported related to TCES, they may use the proportion approach which will make the interpretation easier.
Response 2: Recent literature as well as review paper have been analysed and their information added to the document (introduction). We have counted 139 journals articles published in the last five years that are cited in the paper: that is 64.5 % of the references. The percentage would be higher if we consider conference papers, books and PhD thesis.
Statistic of cited journal papers from the last five years
|
2021 |
2020 |
2019 |
2018 |
2017 |
|
02 |
33 |
39 |
40 |
29 |
Point 3: Please avoid unnecessary punctuation marks including (.) from the end of the title.
Response 3: The comment has been considered when reviewing the whole document. We tried our best to correct this.
Point 4: In the abstract, the authors claimed "Thermochemical energy storage (TCES) emerges as the most promising method to store thermal energy for further use". How thermal energy storage (TES) can store itself? I think it is the material like phase change materials that can participate to store TES.
response 4: The comment has been taking into consideration and the abstract reviewed.
Point 5: Some figures and illustrations are not clear including Graphical Abstract. please avoid unnecessary figure stretching like Figure 6
Response 5: Figures, tables and captions have been reviewed and resized.
Point 6: Surprisingly, the introduction is not supported by a single reference. The introduction is still not enough., need additional information.
Response 6: The introduction has been reviewed and recent references have been listed to supported the assessments.
Point 7: Please thoroughly revise abbreviations, symbols, punctuation, and other required modifications. Meanwhile, please write a similar font style for Figures.
Response 7: Abbreviations and symbols table has been provided after the abstract.
Point 7: The conclusion part needs critical revision, please dig out the main points and outlooks.
Response 7: The conclusion has been reviewed and the outlooks clearly mentioned.

Reviewer 2 Report
Dear Authors,
Although I can see that you put a lot of effort in your paper, I'm not convinced that your paper is adding new information on the table. The things you describe in your paper are written down before by different review articles in the field:
Sarbu, I. and Sebarchievici, C. A Comprehensive Review of Thermal Energy Storage, sustainability, 2018
Scapino, L. et al. Sorption heat storage for long-term low-temperature applications: A review on the advancements at material and prototype scale, Applied energy, 2017
N’Tsoukpoe, A systematic multi-step screening of numerous salt hydrates for low temperature thermochemical energy storage, Applied energy, 2014
N’Tsoukpoe, Kokouvi Edem & Kuznik, Frédéric, 2021. "A reality check on long-term thermochemical heat storage for household applications," Renewable and Sustainable Energy Reviews, Elsevier, vol. 139(C).
In addition, the tables with working temperatures do not mention the conditions of the condenser/evaporator. As experts in the field, you are aware of the fact that this is critical for a good performance of the heat storage device.
In conclusion, I’m missing a new insight in your review article to give a positive recommendation.
Author Response
Response to Reviewer 2 Comments
Although I can see that you put a lot of effort in your paper, I'm not convinced that your paper is adding new information on the table. The things you describe in your paper are written down before by different review articles in the field:
Sarbu, I. and Sebarchievici, C. A Comprehensive Review of Thermal Energy Storage, sustainability, 2018
Scapino, L. et al. Sorption heat storage for long-term low-temperature applications: A review on the advancements at material and prototype scale, Applied energy, 2017
N’Tsoukpoe, A systematic multi-step screening of numerous salt hydrates for low temperature thermochemical energy storage, Applied energy, 2014
N’Tsoukpoe, Kokouvi Edem & Kuznik, Frédéric, 2021. "A reality check on long-term thermochemical heat storage for household applications," Renewable and Sustainable Energy Reviews, Elsevier, vol. 139(C).
In addition, the tables with working temperatures do not mention the conditions of the condenser/evaporator. As experts in the field, you are aware of the fact that this is critical for a good performance of the heat storage device.
In conclusion, I am missing a new insight in your review article to give a positive recommendation
Point 1: New insight in your review article
Response 1: This paper highlights the difficulty of properly addressing the problems encountered in thermochemical heat storage when considering all the disciplines involved in a single paper. It is suggested that thermochemical heat storage be analysed from the microscopic point of view (simulation and analysis of the physics governing the reactions) to the macroscopic point of view (system efficiency and optimization techniques). Besides, the paper suggested a general algorithm and dashboard that can be used for microscopic analysis. The paper recommended a new perspective for future analysis in thermochemical heat storage study.